

# A comparative study of bacterial diversity based on effects of three different shade shed types in the rhizosphere of *Panax quiquefolium* L.

Xianchang Wang[1,*], Xu Guo[1,*], Lijuan Hou[2], Jiaohong Zhang[1], Jing Hu[2], Feng Zhang[1], Jilei Mao[2], Zhifen Wang[1], Congjing Zhang[1], Jinlong Han[1], Yanwei Zhu[1], Chao Liu[1], Jinyue Sun[1] and Chenggang Shan[1]

[1] Key Laboratory of Novel Food Resources Processing, Ministry of Agriculture and Rural Affairs/Research Center of Medicinal Plant, Shandong Academy of Agricultural Sciences/Institute of Agro-Food Science and Technology, Shandong Academy of Agricultural Sciences, Jinan, China
[2] Weihai Academy of Agricultural Sciences, Weihai, China
* These authors contributed equally to this work.

## ABSTRACT

**Background**. Shading is an important factor affecting the cultivation of American ginseng, as it influences crop quality and yield. Rhizosphere microorganisms are also crucial for normal plant growth and development. However, whether different shade types significantly change American ginseng rhizosphere microorganisms is unknown.
**Methods**. This study evaluated the rhizosphere soils of American ginseng under traditional, high flag and high arch shade sheds. High-throughput 16S rRNA gene sequencing determined the change of rhizosphere bacterial communities.
**Results**. The microbial diversity in rhizosphere soils of American ginseng significantly changed under different shading conditions. The bacteria diversity was more abundant in the high arch shade than flat and traditional shades. Different bacterial genera, including *Bradyrhizobium*, *Rhizobium*, *Sphingomonas*, *Streptomyces* and *Nitrospira*, showed significantly different abundances. Different shading conditions changed the microbial metabolic function in the American ginseng rhizosphere soils. The three types of shade sheds had specific enriched functional groups. The abundance of ATP-binding cassette (ABC) transporters consistently increased in the bacterial microbiota. These results help understand the influence of shading systems on the rhizosphere microecology of American ginseng, and contribute to the American ginseng cultivation.

## INTRODUCTION

American ginseng (*Panax quinquefolius* L.) is a perennial understory herb of the Araliaceae family (*Cruse-Sanders & Hamrick, 2004*). The herb improves the overall health of human beings by boosting vitality, improving the immune system and protecting against stress. American ginseng is also a traditional medicine for several pharmacological functions, including anti-inflammatory, anti-cancer activities, anti-diabetes, obesity treatment, and

Corresponding author
Chenggang Shan,
shanchenggang@126.com

enhancing the cardio cerebral vascular, and central nervous systems (*Izzo, 2009*; *Li et al., 2010*; *Poddar et al., 2011*; *Tsao & Liu, 2007*; *Wang, Mehendale & Yuan, 2007*). This herb originated from the eastern part of North America (*Nadeau & Olivier, 2003*) and reached China in the 1980s (*Qin et al., 2018*). Today, the American ginseng planting area in northern China exceeds 10,000 ha (*Jiao et al., 2019*).

In native areas, American ginseng grows as an understory plant in deciduous and mixed forests (*Punja, 2011*), preferring mild and humid climates. The latitude at the origin of American ginseng is similar to vast areas in China, but the rainfall patterns and altitudes vary. This herb prefers higher air humidity with well ventilated, loose soils, suitable for water supply without waterlogging. The optimal growth temperature of American ginseng is 10–28 °C (*Nadeau & Olivier, 2003*). As a typical shade plant, American ginseng prefers oblique, scattered lights. The direct sunlight around noon inhibits photosynthesis, cause photobleaching and leaf death (*Proctor & Palmer, 2017*). Therefore, artificial shades are necessary for American ginseng plantations to mimic natural planting conditions (*Hongpeng et al., 2018*). There are various shade types according to permeability and height. By permeability, the types included, total shade, single and double transparent shades. By height, there are high and low shades. The roof structures may be arch, slope, and flat, constructed using straw cover, wood board, cloth, reed curtain and sunshade net. Shade supporting materials include wood, metal, cement and bamboo pole. The shade structure is classified as simplex and duplex structures, according to the structural complexity. Shades influence the American ginseng soil rhizosphere.

The rhizosphere is the narrow soil region that directly contacts plant roots and facilitates inorganic and organic matter exchange between roots and the soil. This rhizosphere exchange is key for normal plant growth and development (*Broeckling et al., 2008*; *Schmidt et al., 2019*). Plants often alter the rhizosphere for adaptation to changing environments (*Ryan, Delhaize & Jones, 2001*). The rhizosphere harbors numerous microorganisms that constitute a complex community of plant-associated microorganisms important for plant health (*Berendsen, Pieterse & Bakker, 2012*). Microorganisms are essential for plant productivity and health, but some cause plant diseases (*Chisholm et al., 2006*). Plant-associated microorganisms, known as the second genome of plants, have recently attracted extensive research (*Berg et al., 2014*; *Turner, James & Poole, 2013*). American ginseng cultivation is presently the main development agenda, and shading treatment is the decisive factor for ginseng growth and development. Shading determines the economic and biological yield of this vital herb. However, whether or not different shade types change the rhizosphere microorganisms of American ginseng is unknown.

This research analyzed the 16S rRNA bacterial sequences of the American ginseng root rhizosphere to estimate bacterial diversity under three different shade types. The rhizosphere soil has numerous bacterial genera, but the rhizosphere bacteria abundance under the three vary shades. The study results help understand the effects of shading systems on rhizosphere microecology for American ginseng cultivation.

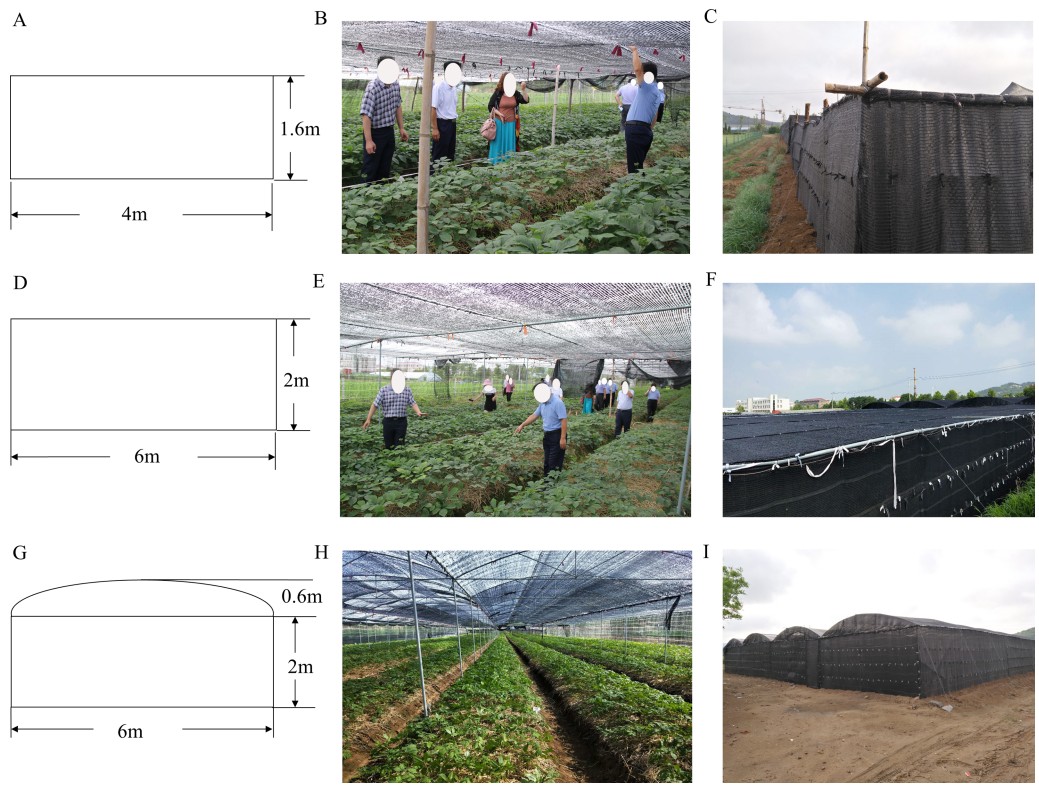

**Figure 1  Schematic drawings and photographs of the three different shade sheds.** (A) Schematic diagram of CTP. (B) A live shot inside the CTP. (C) A live shot of the exterior of the CTP. (D) Schematic diagram of PP. (E) A live shot inside the PP. (F) A live shot of the exterior of the PP. (G) Schematic diagram of GP. (H) A live shot inside the GP. (I) A live shot of the exterior of the GP.

## MATERIALS & METHODS

### General situation of this experiment

The experiment was performed in the Weihai farmland ginseng experimental area of Shandong Province (122°27′E, 36°96′N). Weihai is among the leading American ginseng production areas of China. As a shade-loving plant, American ginseng must be cultivated under shading conditions. Therefore, farmers began to use shade shed to ensure the yield and quality of American ginseng. This study adopted traditional (CTP), high flag (PP), and high arch shade sheds (GP) (Fig. 1). The traditional shed is a flat, wooden structure of 1.6 m, height. The high flat shed is a flat alloy structure, 2.2 m in height, while the high arch shed is an alloy structure measuring 2.6 m in height. All three sheds have double-layer shading nets.

Basal fertilizer (∼3–4 t/mu) was applied once before the experimentation, conducted between 2016 and 2018. The shading shed was a completely random block with three replicates.

## Soil collection

At the end of October 2018 when 3-year-old ginseng roots were manually harvested, 0–10 cm of surface soil was randomly sampled. Six soil samples from one type of shed were combined to form one sample. Each shed type had three composite samples, namely CTP1, CTP2, CTP3, PP1, PP2, PP3, GP1, GP2 and GP3. One part of each sample was stored at −80 °C for subsequent DNA extraction. The other portion was homogenized and sieved with a 2-mm mesh sieve to remove large particles, air-dried and used for chemical analysis. Soil pH was determined at 1:2 soil: water ratio using a combination glass electrode (FE20K, Mettler-Toledo, Swiss). Soil total organic matter (TOM) was assayed following Walkley-Black method (*Walkley & Black, 1934*). Hydrolyzable nitrogen (HN) and available phosphorus (AP) were determined by chemical analytical methods, as described by *Liu et al. (2014)* and *Murphy & Riley (1962)*. The chemical characteristics of these soil samples are shown in Table S1. The yield of American ginseng, disease incidence, type and severity are shown in Table S2. The disease severity index was calculated using the method described by *Chiang et al. (2017)*.

## DNA extraction, PCR amplification and sequencing

Total DNA was extracted using the Z.N.ATM Mag-Bind Soil DNA Kit (Omega Bio-Tek, GA, USA). DNA was separated and visually tested for quality on a 0.8% (w/v) agarose gel electrophoresis. The DNA concentration was measured with a NanoDrop$^{TM}$ 2000 spectrophotometer (Thermo Fisher Scientific, MA, USA). DNA extracts were preserved at −20 °C for PCR.

The bacterial amplicon library was obtained using the 338F (5′-ACTCCTACGGGAGGCA GCA-3′) (*Wu et al., 2016*) and 806R (5′-GGACTACHVGGGTWTCTAAT-3′) (*Wu et al., 2016*) primers, which target the V3–V4 region of the 16S rRNA gene. Personalbio Technology (Shanghai, China) generated 300 bp paired-end reads from the qualified libraries using an Illumina MiSeq platform (Illumina, CA, USA) following the manufacturer's instructions. The raw reads were deposited to the NCBI Sequence Read Archive under accession number PRJNA662686.

## Sequencing data analysis, OTU production and annotation

Split reads were sorted into each sample by their unique barcodes using QIIME V.1.8.0 (*Bokulich et al., 2013*; *Caporaso et al., 2010*). The sequencing barcodes and primer sequences were cut off from original paired-end reads using the FLASH V1.2.7 (*Magoč & Salzberg, 2011*). The USEARCH software (v5.2.236, http://www.drive5.com/usearch/) was used to detect and remove chimera sequences.

The effective tags with ≥97% similarity were assigned to the same operational taxonomic units (OTUs) using the Uparse software (v8.1.1861). The sequence with the highest occurrence frequency in each OTU was considered the representative sequence for further annotation (*Edgar, 2013*). Each representative sequence was annotated using the UCLUST method (*Edgar, 2010*) on the Greengenes database (Release 13.8, http://greengenes.secondgenome.com/) (*DeSantis et al., 2006*). The annotation levels were: kingdom, phylum, class, order, family, genus and species to determine the community

composition of each sample. Rare OTUs (singletons to tripletons), potentially originating from artificial sequences, were removed. The read counts were normalized considering the smallest read number per sample.

## Statistical analysis

All statistical analyses were calculated using the QIIME software (Version 1.8.0) and displayed using the R software (Version 2.15.3). All data were analyzed using a one-way analysis of variance (ANOVA) ($P$ value < 0.05) with the SPSS19.0 software (SPSS Inc., Chicago, USA). The differences between groups were compared using Fisher's least significant difference test.

Alpha diversity was applied to analyze the complexity of species diversity for each community through four indices, including abundance-based coverage (ACE) estimator, Chao 1, Shannon diversity and Inverse Simpson indices. The ACE estimator and Chao 1 index determined the community richness, while the Shannon and Inverse Simpson indices estimated the community diversity. All samples were randomly resampled at the lowest sequencing depth level of 90% to correct the diversity difference caused by sequencing depth. Then, QIIME software is used to calculate alpha diversity.

The principal component analysis (PCA) determined the beta diversity to reduce the dimension of the original variables. Hierarchical clustering was performed using the Unweighted Pair-Group Method with Arithmetic Means (UPGMA) clustering to interpret the distance matrix using average linkage.

The Spearman's rank correlation analysis performed using the Mothur software (*Schloss et al., 2009*). The Cytoscape tool (http://www.cytoscape.org/) displayed the first 50 correlated groups. The PICRUSt tool (http://huttenhower.sph.harvard.edu/galaxy/tool_runner?tool_id=PICRUSt_normalize) determined the enriched pathways following the Kyoto Encyclopedia of Genes and Genomes (KEGG) of the microbiota. The variation analysis was performed following the functional abundance of the samples.

The SPSS 19.0 software determined the Spearman's rank correlation analysis between the microbial community composition and soil physicochemical parameters. Meanwhile, the Redundancy analysis (RDA) was performed using the Canoco 5.0 tool to identify the relationships between the bacterial community structures and soil physicochemical parameters.

# RESULTS

## Sequencing statistic

A total of 660,115 raw tags were generated from 16S rRNA genes of the V3–V4 region. After trimming and filtering, 430,843 high-quality reads were obtained, with a 415 bp average reading length. The OTU clustering (97% similarity) generated 33,858 OTUs from the nine samples, with 99.97% classified under phylum. The rarefaction curves indicated that the sampling sufficiently captured the overall composition of each sample (Fig. S3).

## Taxonomic composition of bacterial assemblages in rhizosphere soil

The taxonomy of 16S rRNA gene amplicons and the relative abundance of the top twenty taxa estimated the bacterial assemblages and variability in American ginseng rhizosphere

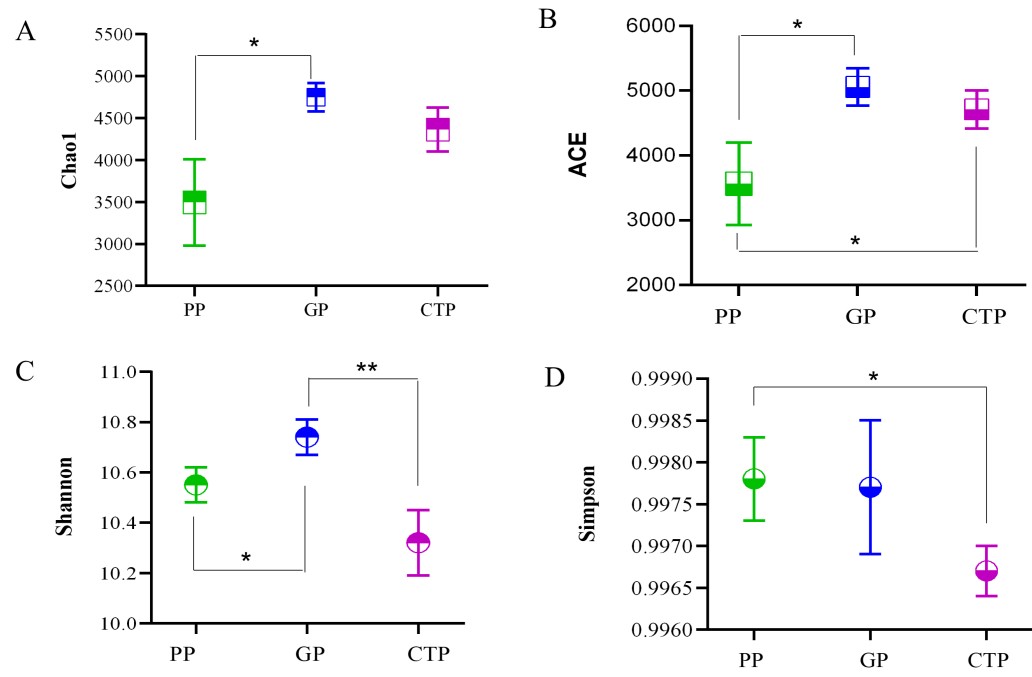

**Figure 2** **The Alpha diversity was estimated by Chao 1, ACE, Shannon index and Simpson in the rhizo-sphere soil under three shade sheds.** Significant changes determined by the T TEST. $P < 0.05$ are marked by an asterisk. $P < 0.01$ are marked by two asterisk. The unmarked are not significantly different.

soils from three different shade sheds (Fig. S4). At the phyla level, Proteobacteria was the most prominent taxon (30.5%) of the total bacterial community, followed by Chloroflexi (17.2%), Acidobacteria (15.4%), Actinobacteria (15.3%), and Gemmatimonadetes (5.8%) (Fig. S4A). The dominant classes with >1% relative abundance across all samples included Alphaproteobacteria, Acidobacteriia, Gammaproteobacteria, KD4-96, Gemmatimonadetes, Bacteroidia, Actinobacteria, Thermoleophilia, Deltaproteobacteria, Subgroup_6, Saccharimonadia, Ktedonobacteria, Anaerolineae, Acidimicrobiia, Clostridia, AD3, Verrucomicrobiae, Planctomycetacia, Blastocatellia (Subgroup_4), and Parcubacteria (Fig. S4B). Bacterial variation increased from phylum to genus, following analysis of the ten most-abundant classes at the order, family and genus levels (Figs. S4C–S4E).

## Alpha and beta diversity analysis of samples

The ACE, Chao 1, Shannon diversity, and Inverse Simpson indices quantified the alpha diversity and per sample bacterial diversities. The ACE and Chao 1 in PP were significantly lower ($P < 0.05$) than GP and CTP (Fig. 2), demonstrating that shading affected the rhizosphere bacterial richness. The Shannon diversity and Inverse Simpson indices showed no significant differences in bacterial diversity among the three shade sheds.

At the OTU level, PC1 explained 60.77% and PC2 13.07% of the total bacterial variation, clearly distinguishing the community structure of soil bacteria under the three shade sheds (Fig. 3A). All the samples collected from the rhizosphere were distinguishable based on the UPGMA-generated cluster trees (Fig. 3B).

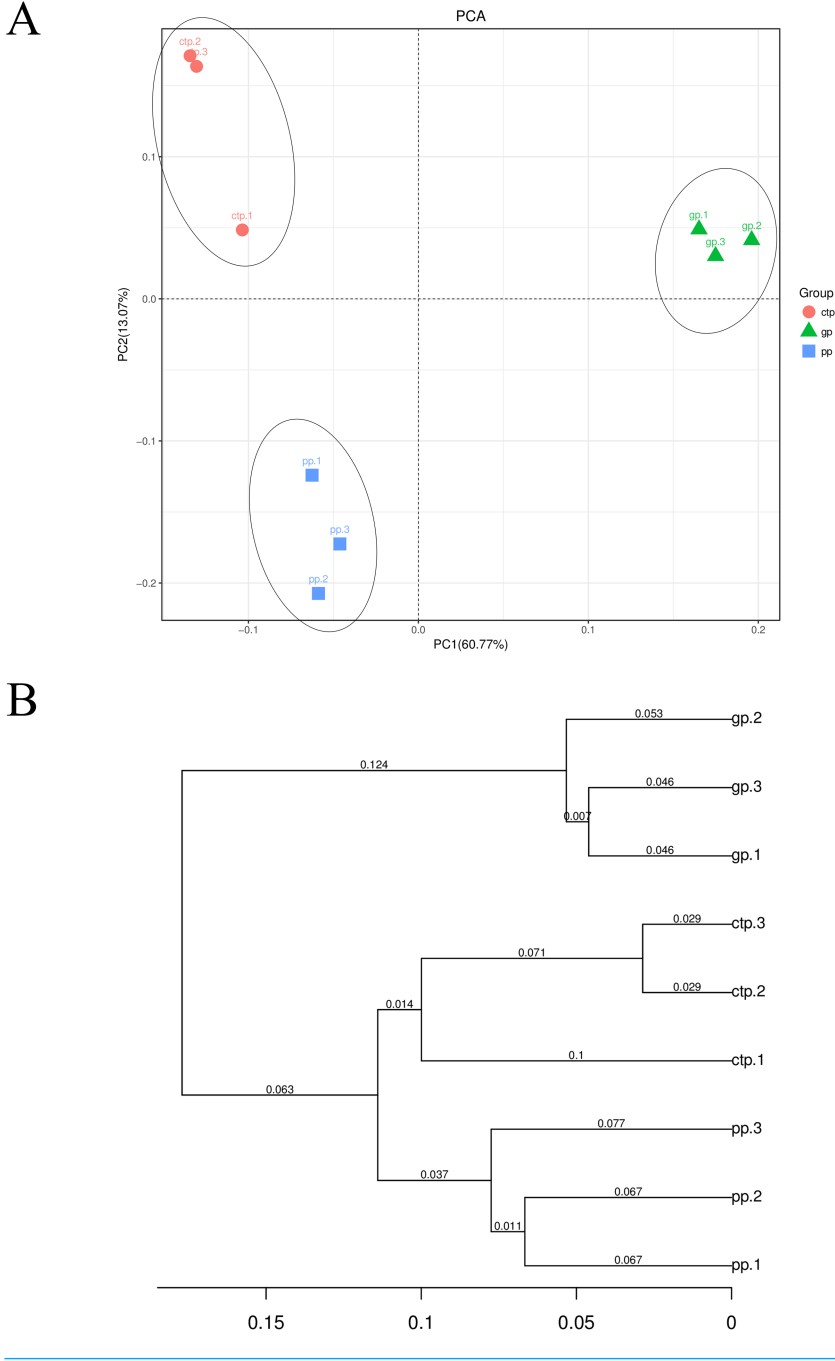

**Figure 3 Beta diversity of the bacterial communities.** (A) PCA analysis of the soil bacterial communities based on the relative abundance of OTU. (B) UPGMA cluster analysis based on Weighted UniFrac.

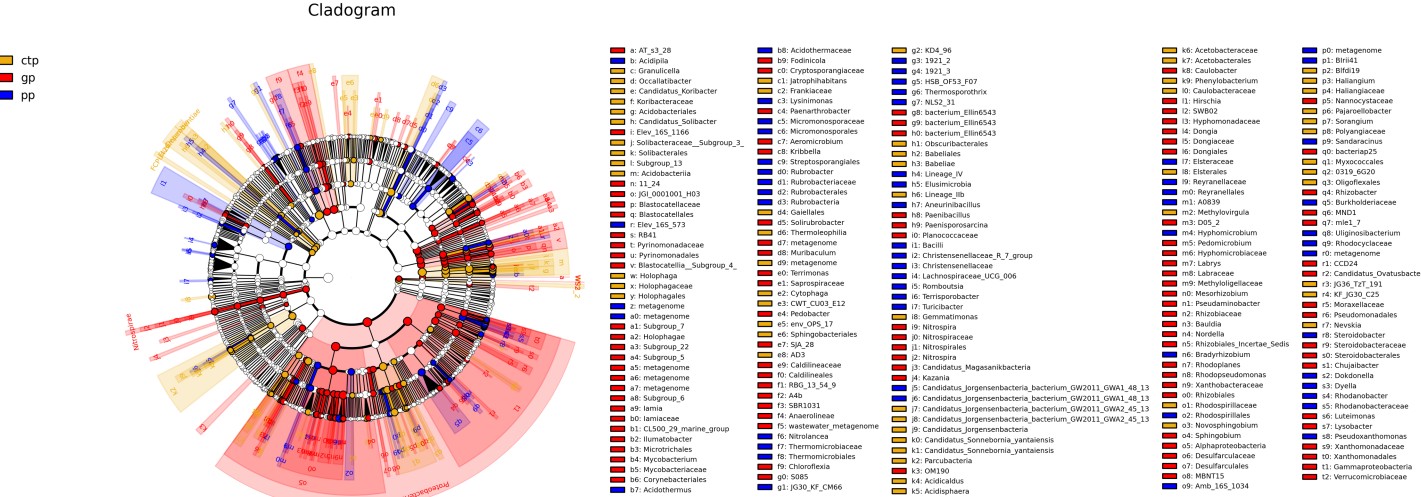

**Figure 4** **The inter-group difference classification unit display map based on the classification hierarchy tree.** The taxa hierarchy of all from gate to genus is shown from the inner to the outer circle. Node size represents the average relative abundance of OTUs; white nodes are OTUs with no significant difference between groups; red nodes are OTUs with high abundance in GP; yellow nodes are OTUs with high abundance in CTP; blue nodes are OTUs with high abundance in PP; letters are OTUs names with significant differences between groups.

## Variations in the bacterial community of American ginseng rhizosphere soil under three different shade sheds

The least discriminant analysis (LDA) effect size taxonomic cladogram compared the phylum to the genus in the three sample communities to expose the bacterial variation in the American ginseng rhizosphere soil under three different shade sheds. The relative abundances of bacterial phyla were different across the three shade shed rhizosphere soils (Fig. 4). The relative abundances of Proteobacteria, Nitrospirae and WS2 in GP were significantly higher than PP and CTP (LSD, $p < 0.05$). Dependentiae, Elusimcrobia, FCPU426 and WPS-2 were significantly abundant in the pp than GP and CTP (LSD, $p < 0.05$). At a finer taxonomic level, the three shade sheds were significantly different in some bacterial genera. The abundance values of the 50 most abundant genera from the three shade sheds are shown in Fig. S5.

## Correlation and KEGG functional analysis in bacterial communities

The relationships between abundance values of the various bacterial genera were determined following the abundance changes of different genera in the soil samples. The genera relationship network indicated a complex functional collaboration within the microbiota (Fig. S6). The KEGG functional enrichment analysis of bacterial microbiota from rhizosphere soils under the three different shade sheds determined their bacterial functional diversities. The abundance of the top 50 functional groups was analyzed and displayed on a heat-map (Fig. 5). A Venn diagram (Fig. 6) shows the number of OTUs of the common functional soil bacteria in the three shade sheds, indicating the functional divergence of bacterial microbiota in the rhizosphere of American ginseng roots.

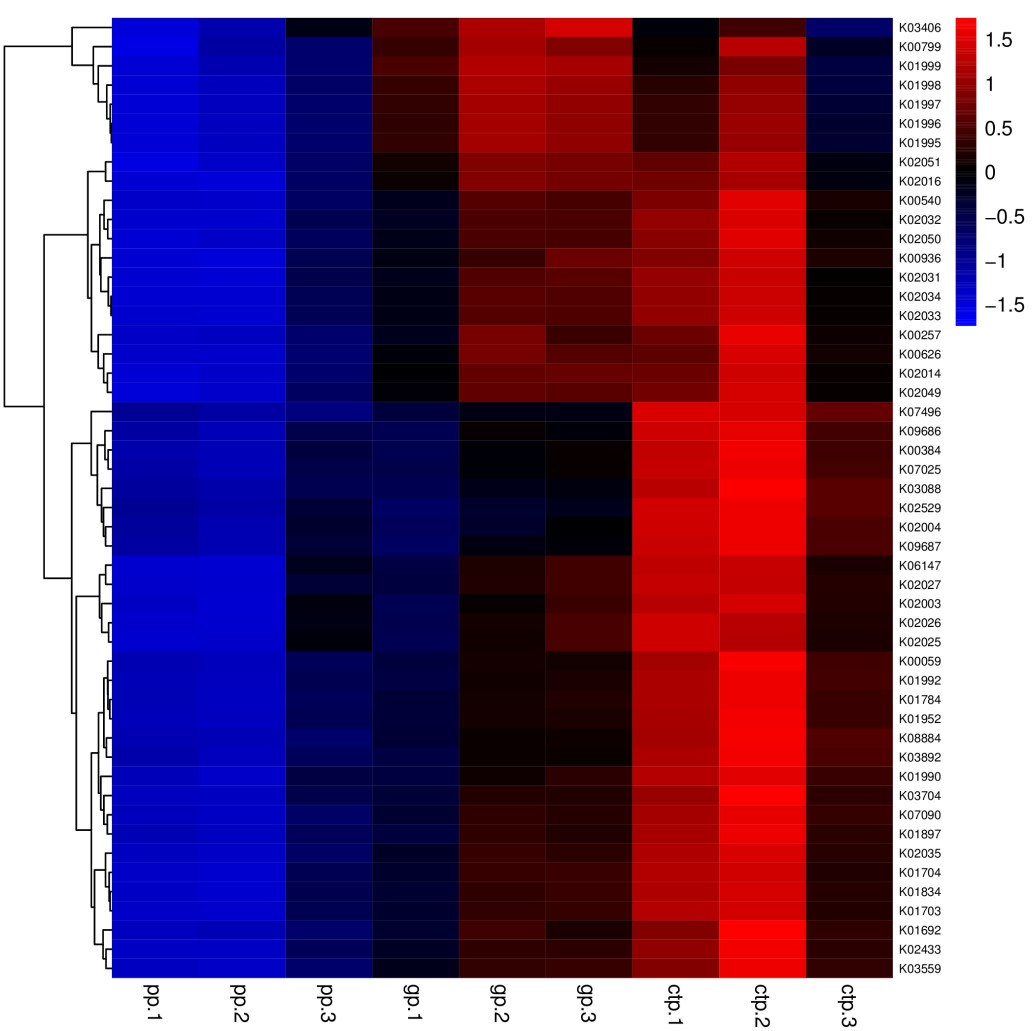

**Figure 5** **Heatmaps of the top 50 enriched functional groups.** The color code refers to gene abundance, with high predicted abundances (red) and low predicted (blue).

## Relationship between microbial communities and soil chemical properties

The Spearman's rank correlation coefficient determined the relationship between soil microbial diversity and soil chemical properties (Fig. 7). The Chao 1 and Shannon diversity indices positively and significantly correlated with the soil pH ($P < 0.05$). However, Chao 1 and ACE indices negatively correlated with soil TOM and AP contents ($P < 0.01$), while the Inverse Simpson index negatively correlated with HN ($P < 0.05$).

RDA analysis was performed to observe the potential correlations between the community structures at genus levels and soil chemical properties in the GP, PP and CTP. The RDA analysis results showed that the first two RDA axes (RDA1 and RDA2) account for 60.30% and 12.67% of the variance, respectively (Fig. 8). Soil pH ($F = 9.2$, $P = 0.016$) was the most important contributor to bacterial community variation (Table S7).
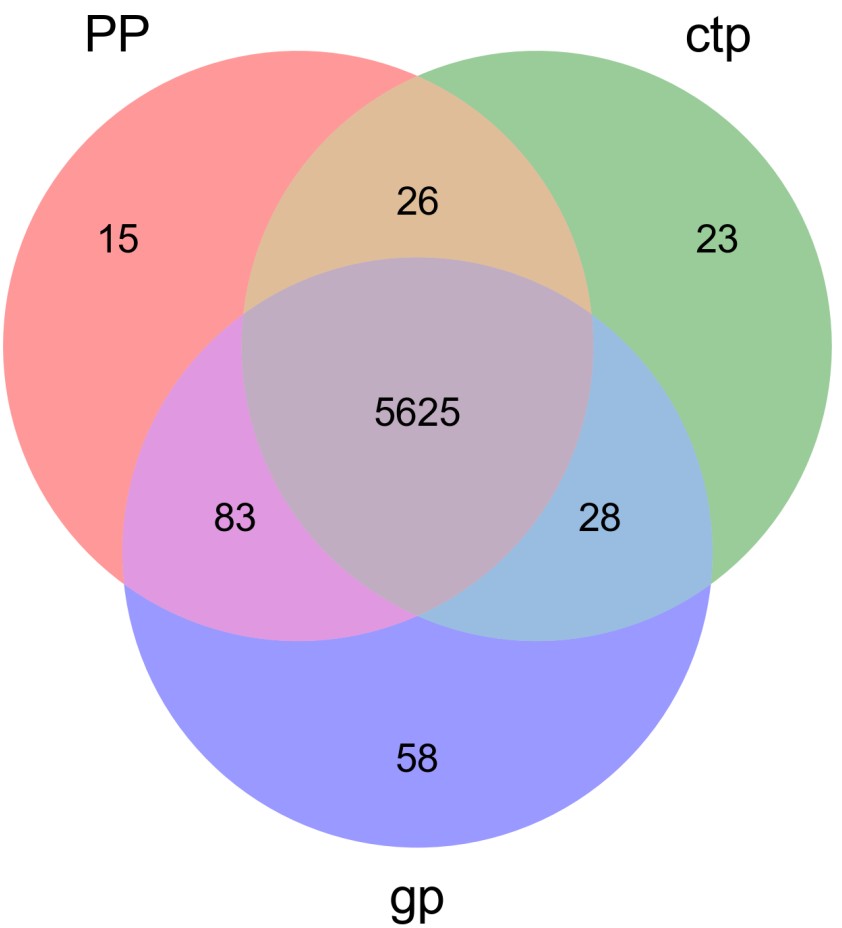

**Figure 6** Venn diagram of OTUs for common functionally enriched soil bacteria among the three different shade shades.

## DISCUSSION

This study investigated the changes of American ginseng rhizosphere soil bacterial communities under three different shade sheds. The GP microbial diversity was higher than PP and CTP. Meanwhile, the soil physicochemical properties affected microbial diversity. It has been reported that the bacterial diversity of grapes rhizosphere changed under different environmental conditions (*Bokulich et al., 2014*). In this study, the soil physicochemical properties under the different shade sheds were different (Table S1). The pH of GP was significantly higher than PP and CTP, while the TOM and AP content of PP was higher than GP and CTP. pH is a key driving factor of the soil bacterial community, which determines the availability of soil nutrients (*Lauber et al., 2009*; *Shi et al., 2015*). Increasing the pH increased the soil bacteria diversity (*Rousk, Brookes & Bååth, 2009*). In this study, Spearman's correlation analysis showed that the pH positively and significantly correlated with the microbial diversity (Fig. 7).

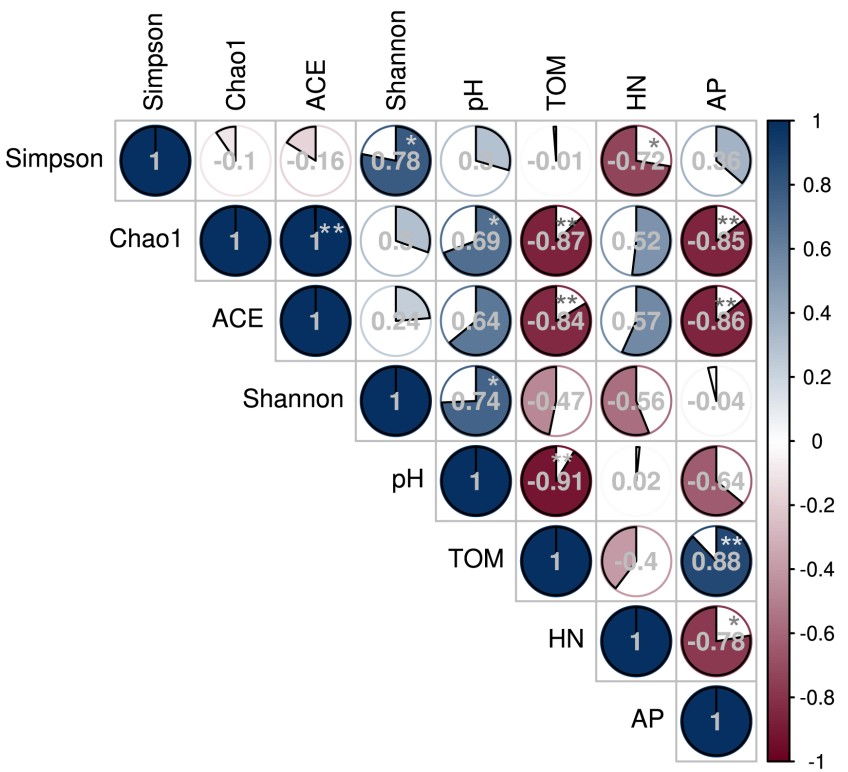

**Figure 7 Correlation between soil bacterial diversity index and physicochemical properties under different shade sheds.**

Besides the above attributes, the most critical resource for bacterial growth is organic matter (*Laganière et al., 2013*), and nitrogen is the second limiting nutrient (*Frey, 2004*). However, TOM, AP and HN negatively correlated with microbial diversity. The community structure of microorganisms varied under the three different shade sheds (*Van der Heijden & Wagg, 2013*). The soil physicochemical properties affected the microbial structure, shown by RDA analysis. However, pH was the key environmental factor affecting the composition of the microbial community (Fig. 8). The diversity of soil microorganisms is related to soil health and quality (*Anderson, 2003*; *Garbeva, Van Veen & Van Elsas, 2004*). Decreased soil microbial diversity causes soil-borne plant diseases while changing growth conditions increase the risks of phytopathogens (*Burie, Langlais & Calonnec, 2011*; *Pugliese, Gullino & Garibaldi, 2011*).

This study reported varying incidences and severity of diseases affecting American ginseng under the three shade types. The average disease incidence of American ginseng under GP, PP and CTP were 3.58, 5.19, and 12.65%, respectively. However, the average disease severity for GP, PP and CTP were 10.5, 9.8, and 18.6%, respectively. The incidence and severity of American ginseng diseases under GP and PP were significantly lower than CTP. Interestingly, the fresh weight of American ginseng in GP and PP was higher than in CTP. Therefore, the type of shade is critical for disease control and the yield increase in American ginseng.

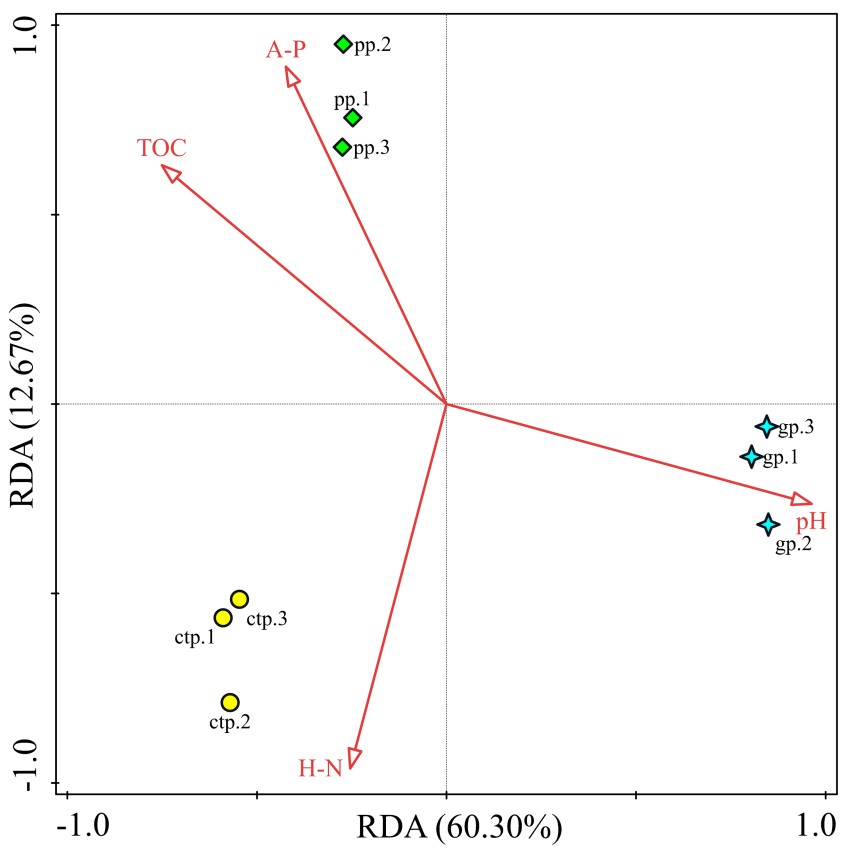

**Figure 8** The RDA relationship between soil physicochemical and bacterial community composition.

American ginseng is a perennial plant whose root exudates accumulate in the rhizosphere, providing substrates for several biological communities. Root exudates are considered a driving force for selecting specific microbial populations in the rhizosphere (*Bais et al., 2006*; *Garbeva, Van Veen & Van Elsas, 2004*; *Mazzola & Manici, 2012*). For example, the proportion of Proteobacteria and Bacteroidetes increased because of their increased relative abundances in a high nitrogen area (*Fierer et al., 2012*). Gram-negative Proteobacteria are dominant in the three soil samples used in this study, causing legume symbiotic nitrogen fixation (*Raymond et al., 2004*). *Bradyrhizobium*, a gram-negative Proteobacteria, had the highest relative abundance in the rhizosphere soil of the American ginseng sampled in this study. *Bradyrhizobium*, genus Azotobacterthat, induces nodule formation in the roots of legumes (*Long, 1996*). However, the American ginseng lacked root nodules, probably due to the absence of nod receptors or defects of subsequent kinase cascades in American ginseng, which are essential for legume nodule formation (*Gage, 2004*; *Smit et al., 2007*). *Sphingomonas* was the second largest genus in the rhizosphere of American ginseng under the three shade types. Genus *Sphingomonas* commonly degrade various aromatic compounds (*Fredrickson et al., 1995*). Therefore, *Sphingomonas* accumulation in the rhizosphere indirectly implies the secretion of different aromatic secondary metabolites in American ginseng roots. Another abundant genus, *Nitrospira* species, is critical for the

nitrogen cycle in water and soil (*Bartosch et al., 2002*). The content of both *Sphingomonas* and *Nitrospira* in the high arch shed soil was higher than the other two sheds. The study also identified that *Gemmatimonas* had high relative abundance. The *Gemmatimonas* acquire various plant processing resources, and their abundance negatively correlates with plant growth (*Franke-Whittle, Manici & Heribert Insam, 2015*).

As expected, the relative abundance of some microbial communities decreased due to specialized antimicrobial metabolites in the root exudates (*Bais et al., 2006*; *Berg & Smalla, 2009*; *Mazzola & Manici, 2012*). Additionally, changes in the chemical properties of soils probably changed the microbial community composition (*Lauber et al., 2008*). The analyses showed significantly different pH, organic matter, hydrolytic nitrogen, and available phosphorus among the three different shade sheds (Table S1), but the effect mechanism on the microbial community needs further verification. Changes in the composition of bacterial communities may change metabolism, biodegradation and disease inhibition (*Bell et al., 2013*; *Garbeva, Van Veen & Van Elsas, 2004*). These results show that shade sheds improve soil productivity, and appropriate shade shads can modify soil microbial communities.

Considering the KEGG enrichment analysis, half of the 50 most abundant functional groups in the microbial community of the three shade types are transporters (Fig. 5). The other half involved metabolism, such as amino acid (*e.g.*, k00384, k01704, k01703), fatty acid (*e.g.*, k00626, k00059, k01897) and secondary metabolism (*e.g.*, k01952, k01834, k01692). Among the 26 transporters, 22 are ATP-binding cassette (ABC) transporters. ABC transporters are integral membrane proteins that combine substrate trans lipid bilayers transport with ATP hydrolysis (*Hollenstein, Dawson & Locher, 2007*). ABC transporters are essential for the survival of bacteria as they catalyze nutrient uptake by bacteria and facilitate the efflux of toxic or antibacterial drugs. ABC transporters also export various virulence factors, including antibiotics, bacteriocins, and toxins, increasing within-community competition and hindering pathogen invasion (*Davidson & Chen, 2004*). The accumulation of bacterial rich in ABC transporters in the rhizosphere mediates inorganic and organic matter exchange between the root and soil (*Ryan, Delhaize & Jones, 2001*). CTP had the most abundant expression of ABC transporters (Fig. 5), probably because CTP had the highest pathogenic rate and required more ABC transporters for defense against pathogen infection. The structure and composition of microbial communities are closely related to their metabolic function. This study identified 5625 functional groups in the three shade types, including 15 specific functional groups for the flat shed, 23 for the traditional shed, and 58 for the high arch shed (Fig. 6). The high arch shade sheds had a significantly higher number of specific functional groups than the other two sheds.

## CONCLUSIONS

The diversity of American ginseng soil microbial community structure and function changed under the three different shade sheds. Shade sheds improve soil productivity, an effective defense mechanism for American ginseng. Shade sheds also balance soil microbial communities, and shading enriches the metabolic functions of soil microorganisms.

This work provides a theoretical understanding of shading effect on the rhizosphere microecology of American ginseng.

### Funding
This study was supported by the Agricultural Science and Technology Innovation Project of Shandong Academy of Agricultural Sciences, China (NO. CXGC2021A50, CXGC2021A18); the National Natural Science Foundation of China (No. 82003632); the Provincial Major Scientific and Technological Innovation Project of Shandong, China (NO. 2019JZZY020612); the Province Agricultural Major Application Technology Innovation Project of Shandong, China (NO. SD2019ZZ016); the Provincial Key Research and Development Program of Shandong, China (NO. 2019LYXZ025, 2019GSF109087); the Taishan Scholars's Program of Shandong for Jinyue Sun. The funders had no role in study design, data collection and analysis, decision to publish, or preparation of the manuscript.

### Grant Disclosures
The following grant information was disclosed by the authors:
Agricultural Science and Technology Innovation Project of Shandong Academy of Agricultural Sciences, China: CXGC2021A50, CXGC2021A18.
The National Natural Science Foundation of China: 82003632.
The Provincial Major Scientific and Technological Innovation Project of Shandong, China: 2019JZZY020612.
The Province Agricultural Major Application Technology Innovation Project of Shandong, China: SD2019ZZ016.
The Provincial Key Research and Development Program of Shandong, China: 2019LYXZ025, 2019GSF109087.
The Taishan Scholars's Program of Shandong.

### Competing Interests
The authors declare there are no competing interests.

### Author Contributions
- Xianchang Wang and Xu Guo analyzed the data, prepared figures and/or tables, authored or reviewed drafts of the paper, and approved the final draft.
- Lijuan Hou performed the experiments, prepared figures and/or tables, authored or reviewed drafts of the paper, and approved the final draft.
- Jiaohong Zhang, Jing Hu, Feng Zhang, Jilei Mao, Zhifen Wang, Congjing Zhang, Jinlong Han and Yanwei Zhu performed the experiments, authored or reviewed drafts of the paper, and approved the final draft.
- Chao Liu and Jinyue Sun conceived and designed the experiments, authored or reviewed drafts of the paper, and approved the final draft.
- Chenggang Shan conceived and designed the experiments, analyzed the data, prepared figures and/or tables, authored or reviewed drafts of the paper, and approved the final draft.

## Data Availability

GenBank: PRJNA662686

## Supplemental Information

Supplemental information for this article can be found online at http://dx.doi.org/10.7717/peerj.12807#supplemental-information.

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
