# Peer review of "A comparative study of bacterial diversity based on effects of three different shade shed types in the rhizosphere of Panax quiquefolium L"

_PeerJ, doi:10.7717/peerj.12807_

## Round 0.1 · original submission · Major Revisions

Please, address all the comments of the reviewers and carefully revise the language of the article as both reviewers suggested the help of a professional English editor for its revision. Also, the improvement of the results and discussion sections is required, together with the consideration of the individual conclusions.

·

Basic reporting

The paper use clear, unambiguous, technically correct text. Most part of he article conform to professional standards of courtesy and expression,however,you‘d better find a professional native language English speakers to help revise the full article.
The paper include sufficient introduction and background to demonstrate how the work fits into the broader field of knowledge.
The structure of the article basically conform to an acceptable format of ‘standard sections’.
Figures in this paper is relevant to the content of the paper, of sufficient resolution, and appropriately described and labeled.

Experimental design

The submission clearly define the research question, which was relevant and meaningful. The knowledge gap being investigated was identified clearly.

Methods described in this article had sufficient information to be reproducible by another investigator.

Validity of the findings

The data on which the conclusions are based made available in an acceptable discipline-specific repository. The data was robust, statistically sound, and controlled.

Conclusions are well stated, linked to original research question. However, Individual conclusions need to be carefully considered.

Additional comments

Overall, I agree that this article will be published in Peer J unless you modify your paper according to senior editor and my suggestion. Suggestion will be upload in the specified page.
you‘d better find a professional native language English speakers to help revise the full article.

Reviewer 2 ·

Basic reporting

The language and grammar of the article including the title requires revisions.

Experimental design

Line 95-102 The description of climate characteristics in Weihai area is too detailed, please simplify it.
Line 114-115 The sampling information is too simple, please describe it in detail. Please specify the number of specific sampling sites.
Line 118 Please describe the chemical analysis in detail and cite related references.
Line 200 Fig. S6C–S6E? Please check it.
Line 234 ginkgo roots?

Validity of the findings

The difference in environmental conditions and fresh weight, disease rate etc. of American ginseng under three different shade sheds should be analyzed in Results and Discussion parts.
Were the soil properties the same in each experiment site before American ginseng was planted? Did the type of shade sheds cause the difference in soil properties? Please explain them in Results and Discussion parts.
The analysis of the reason (how shed type affected soil properties) is lacking in the Discussion part.

---

## Round 0.2 · accepted · Accept

The reviewers agree to accept the current reviewed version of your manuscript.

Reviewer 2 ·

Basic reporting

No comment

Experimental design

No comment

Validity of the findings

No comment

Additional comments

No comment